# Characterization of the Gene Expression Profile Response to Drought Stress in *Populus ussuriensis* Using PacBio SMRT and Illumina Sequencing

**DOI:** 10.3390/ijms23073840

**Published:** 2022-03-30

**Authors:** Wenlong Li, Zhiwei Liu, He Feng, Jingli Yang, Chenghao Li

**Affiliations:** State Key Laboratory of Forest Genetics and Tree Breeding, Northeast Forestry University, 26 Hexing Road, Harbin 150040, China; liwenlong425@nefu.edu.cn (W.L.); 2021111160@nefu.edu.cn (Z.L.); 2020023050@nefu.edu.cn (H.F.); yifan85831647@nefu.edu.cn (J.Y.)

**Keywords:** drought stress, *Populus ussuriensis*, root, single-molecule real-time sequencing, alternative splicing

## Abstract

In this study, we characterized the gene expression profile in the roots of *Populus ussuriensis* at 0, 6, 12, 24, 48 and 120 h after the start of polyethylene glycol (PEG)-induced drought stress using PacBio single-molecule real-time sequencing (SMRT-seq) and Illumina RNA sequencing. Compared to the control, 2244 differentially expressed genes (DEGs) were identified, and many of these DEGs were associated with the signal transduction, antioxidant system, ion accumulation and drought-inducing proteins. Changes in certain physiological and biochemical indexes, such as antioxidant activity and the contents of Ca^2+^, proline, and total soluble sugars, were further confirmed in *P. ussuriensis* roots. Furthermore, most of the differentially expressed transcription factors were members of the AP2/ERF, C2H2, MYB, NAC, C2C2 and WRKY families. Additionally, based on PacBio SMRT-seq results, 5955 long non-coding RNAs and 700 alternative splicing events were identified. Our results provide a global view of the gene expression profile that contributes to drought resistance in *P. ussuriensis* and meaningful information for genetic engineering research in the future.

## 1. Introduction

Drought is one of the major natural disasters facing plants. In recent years, due to the influence of the global climate, the frequency of droughts has been increasing [1]. In plants, the root is the first organ to sense drought stress, and it plays a vital role in the response of plants to drought stress [2]. Hence, the roots are important in maintaining a good growth in plants, especially in soils with insufficient moisture or nutrients. During water deficit, the root perceives the soil water status, promotes biosynthesis and signal transduction of abscisic acid (ABA), and promotes changes in the expression level of drought-related genes [3,4].

Plants are sessile organisms with complex mechanisms to cope with drought stress. Strategies for dealing with drought stress in plants include (1) changes in the structure of the root system at the phenotypic level, (2) changes in osmotic regulation and antioxidant metabolism at the physiological level, and (3) regulation of drought signaling, transcription factors (TFs), and cellular defense at the molecular level [5,6,7]. Studies have exhaustively investigated molecular responses to drought stress in plants, such as *Populus davidiana* [8], *Gossypium hirsutum* and *Gossypium tomentosum* [9], *Arabidopsis thaliana* [10], and *Oryza sativa* [11]. TFs play an important role in the molecular response mechanism under drought stress, such as ABA-responsive element (ABRE), myeloblastosis (MYB), ethylene-responsive factor (ERF) and WRKY [12,13,14]. Additionally, synthesis of certain specific proteins, such as heat shock protein (HSP) and late embryogenesis abundant (LEA) protein, enhances the drought tolerance of plants [15,16]. However, the gene network involved in the response to drought stress in plants has not yet been fully elucidated. A further understanding of the drought regulation mechanism will provide theoretical support for the development of new drought-resistant plant varieties.

*Populus ussuriensis*, a member of *Populus* sect., is widely used for the regeneration of mountain forests in regions of Northeast China. The model of *P. ussuriensis* in response to drought stress remains unexplored. At present, there are no reports on whole-genome sequencing of *P. ussuriensis*, which limits the study of distinct molecular mechanisms in the plant. The PacBio single-molecule real-time sequencing (SMRT-seq) can countervail these limitations. The full-length transcripts obtained using PacBio SMRT-seq can be used directly for alternative splicing (AS) events identification, simple sequence repeats (SSRs) identification, and long noncoding RNA (lncRNA) prediction.

In this study, the roots of *P. ussuriensis* were used as the research material, and PacBio SMRT-seq and Illumina RNA-seq were used to explore the physiological and molecular changes in *P. ussuriensis* under polyethylene glycol (PEG)-induced drought stress. We aimed to determine the gene regulation network involved in the adaptation of *P. ussuriensis* to PEG-induced drought stress to highlight candidate genes that can be used to enhance drought resistance of *P. ussuriensis* and other plant species.

## 2. Results

### 2.1. Overview of the PacBio SMRT-seq and Illumina RNA-seq

In this study, 18 samples were mixed to construct a SMRTbell library for obtaining comprehensive transcriptome profiles of *P. ussuriensis* roots under PEG-induced drought stress. In total, 31,547,319 subreads (average length of 943 b.p. and N50 length of 1435 b.p.) and 387,340 circular consensus sequencing (CCS) reads (average length of 1461 b.p. and N50 length of 2133 b.p.) were obtained after quality control (Table 1). A total of 40,307 high-quality (HQ) isoforms with an average length of 1407 b.p. and 152 low-quality (LQ) isoforms with an average length of 847 b.p. after classification and correction of CCS reads. As shown in Figure 1A,B, the length distribution of CCS reads and consensus isoforms was predominantly between 500 and 4000 b.p.

For Illumina RNA-seq, a total of 816,789,418 clean reads were obtained, with a clean Q30 base rate > 93% for all samples. Then, the clean reads were successfully mapped back to the full-length transcripts of PacBio SMRT-seq. As shown in Appendix A, the average mapping ratio is 69.3%, the average multimap ratio is 28.8%.

### 2.2. SSR and lncRNA Prediction

In this study, 3121 SSRs were identified using the MISA software (Appendix A). As shown in Figure 1C, the type distribution included 1252 dinucleotides (40.1%), 1673 trinucleotides (53.6%), and 196 tetranucleotides. As shown in Figure 1D, Coding-non-Coding Index (CNCI), Coding Potential Assessment Tool (CPAT), and Coding Potential Calculator (CPC) software predicted 11,484, 11,294, and 8481 lncRNAs, respectively.

### 2.3. Identification of Alternative Splicing

In total, 700 AS events were identified using the AStalavista tool (Appendix A). As shown in Figure 2A, the majority of AS events were of retained intron (RI), alternative 3’ splice site (A3SS), or alternative 5’ splice site (A5SS) types. Gene ontology (GO) enrichment analysis was performed on AS genes; the top 10 functional groups of biological process (BP), molecular function (MF), and cellular component (CC) categories are shown in Appendix A. The top three GO enrichment results for BP category were “transcription”, “regulation of transcription” and “mRNA processing”, and the top three for MF category were “ATP binding”, “metal ion binding” and “DNA binding”. As shown in Figure 2B, 281 AS genes were mapped to the 37 pathways in the Kyoto Encyclopedia of Genes and Genomes (KEGG) database.

### 2.4. Transcript Integrity Analysis and Gene Annotation

The HQ isoforms generated by PacBio SMRT-seq were used to open reading frames (ORFs) identification after integrity assessment using single-copy orthologs (BUSCO). A total of 34,522 ORFs (297–6624 b.p. long) were identified using TransDecoder. The average length of ORFs is 1134 b.p., and the N50 of ORFs is 1473 b.p. (Figure 2C). After removing the redundancy of ORFs, 14,136 genes were annotated using Trinotate (Appendix A). For further functional prediction and categorization, 8967 genes were annotated using the euKaryotic Orthologous Groups (KOG) method (Appendix A). For GO enrichment analysis, 10,692 ORFs were successfully enriched into BP (9211), CC (9383), and MF (9290) categories (Appendix A).

### 2.5. Identification and Analysis of DEGs

The differentially expressed genes (DEGs) were determined using the DEGseq software, which is based on the read count values of each transcript. In total, 2244 DEGs (1328 DEGs were up-regulated and 959 DEGs were down-regulated) were identified under PEG-induced drought stress (Figure 3B, Appendix A). Our results demonstrated that 871 (65.59%) and 786 (59.19%) DEGs were up-regulated at 6 or 12 h, respectively, while 679 (70.8%) and 429 (44.73%) DEGs were down-regulated at 6 or 12 h, respectively (Figure 3B). In the down-regulated category, only 29 genes showed decreased expression at all time points, and in the up-regulated category, only 89 genes showed decreased expression at all time points. A heatmap, which can visualize the changing pattern of DEGs under different time points of PEG treatment was constructed based on the read count values (Figure 3A).

In GO enrichment analysis of all DEGs, 1607, 1604, and 1520 DEGs were enriched in the BP, MF, and CC categories, respectively (Appendix A). A total of 662 DEGs were mapped to 134 KEGG pathways, and the metabolic pathways regulated by PEG-induced drought stress were screened out (*p* < 0.05; Figure 4). Our results demonstrated that metabolic pathways related to drought stress were more significantly enriched in the early stages (6, 12, and 24 h) than in the late stages (48 and 120 h). The phenylpropanoid biosynthesis pathway was significantly enriched at all time points, and the MAPK signaling, flavonoid biosynthesis, and amino acid metabolism pathways were significantly enriched at most of the time points.

### 2.6. DEGs Responding to Drought Stress in P. ussuriensis

DEGs associated with the response to drought stress in plants are mainly involved in drought signal transduction and defense system activation. All DEGs responding to drought stress are shown in Appendix A, and some of these are listed in Table 2. The perception and transmission of drought signals by *P. ussuriensis* plays a crucial role in the response to drought stress. In this study, we identified 38 DEGs involved in the transmission of drought signals in *P. ussuriensis* (Appendix A). Of these, 14 DEGs, including two 9-*cis*-epoxycarotenoid dioxygenase (NCED) homologous genes, four PYR1-like (PYL) homologous genes, and two protein phosphatase 2C (PP2C) homologous genes, were involved in ABA synthesis and binding. Of the total 38 DEGs, 20 DEGs, including calmodulin-like (CML), calmodulin-binding protein, and calcium-dependent protein kinase (CPK/CDPK) homologous genes, were significantly differentially expressed under drought stress. In addition, we found that four mitogen-activated protein kinase kinase kinase (MAPKKK) homologous genes were significantly up-regulated under PEG-induced drought stress.

Furthermore, 56 DEGs related to reactive oxygen species (ROS) scavenging were identified under PEG-induced drought stress, including those encoding catalase (CAT), glutathione S-transferase (GST), peroxidase (POD) and superoxide dismutase (SOD) (Appendix A). Of these, most of the GST homologous genes were significantly up-regulated after 6, 12, and 24 h of PEG-induced drought stress. In *P. ussuriensis*, a total of 28 POD homologous genes were also found to be significantly differentially expressed under PEG-induced drought stress.

After PEG treatment, 13 DEGs related to ion transport were found in the roots of *P. ussuriensis*. The expression levels of two genes (*f7p60_2856_19557* and *f2p60_2781_19668*) encoding potassium transporter (KT) homologous genes, one gene (*f2p37_2396_20662*) encoding cyclic nucleotide-gated ion channel (CNGC) homologous gene, two genes (*f2p55_3201_1361* and *f2p60_3228_18926*) encoding glutamate receptor (GLR) homologous genes, two genes (*f3p60_1980_22126* and *f4p60_1851_22830*) encoding cation exchanger (CAX) homologous genes, and two genes (*f2p57_4009_433* and *f2p20_3494_18605*) encoding calcium-transporting ATPase (ACA) homologous genes were significantly different.

In this study, 38 DEGs encoding HSP homologous genes and 19 DEGs encoding LEA homologous genes were identified under PEG-induced drought stress (Appendix A). Of these, most of the DEGs were significantly up-regulated at 6, 12, and 24 h of PEG-induced drought stress. Moreover, five DEGs encoding dehydrin family proteins were up-regulated at all time points. Our results identified 24 DEGs related to the lignin biosynthetic process in *P. ussuriensis*, including 4-coumarate-CoA ligase (4CL), *trans*-cinnamate 4-monooxygenase (C4H), cinnamyl alcohol dehydrogenase (CAD), cinnamoyl CoA reductase (CCR), caffeic acid 3-O-methyltransferase (COMT), and phenylalanine ammonia lyase (PAL) (Appendix A). In addition, more DEGs were identified at 6, 12, and 24 h of PEG-induced drought stress.

### 2.7. Transcription Factors Responding to Drought Stress in P. ussuriensis

According to reports, many TFs are affected by drought conditions. During PEG-induced drought stress, 122 TFs were differentially expressed, which belonged to 18 different families (Appendix A). Of these, 10 TF families accounted for 86% of stress response, including AP2/ERF, C2H2, MYB, NAC, C2C2, WRKY, HB, bHLH, AUX/IAA, and C3H (Figure 5A). A heatmap was constructed for visualizing the change of TFs at all time points of the PEG-induced drought stress treatment (Figure 5B). The number of TFs was greater at 6 and 12 h than that at 48 and 120 h, probably because many TFs were activated in the early stages of drought stress. As shown in Figure 5C, we selected several TFs (e.g., TFs encoding ERF017, ZAT11, HB7, MYB58, NAC35, and WRKY75 homologous genes) to conduct quantitative reverse transcription-polymerase chain reaction (qRT-PCR) analysis.

### 2.8. Validation of Gene Expression Levels

The results of qRT-PCR analysis of 12 genes were consistent with that of Illumina RNA-seq data (Figure 6), indicating the reliability of the Illumina RNA-seq data.

### 2.9. Changes in Physiological Indexes in Response to Drought Stress

We measured the H_2_O_2_ content and the activities of GST, SOD, and POD for evaluating the drought-induced phenotypes. After PEG-induced drought stress, from 6 to 48 h, the H_2_O_2_ content levels in *P. ussuriensis* were significantly increased; however, the levels dropped significantly at 120 h (Figure 7A). POD, SOD, CAT, and GST activities showed a similar trend after PEG-induced drought stress by significantly increasing at 6, 12, 24, 48, and 120 h and reaching the highest at 24 h (Figure 7B–E).

After PEG-induced drought stress, the proline content levels in *P. ussuriensis* significantly increased at 24, 48, and 120 h (Figure 7F). The total soluble sugar content exhibited a significant increase during PEG-induced drought stress treatment and was maintained at a high level at 12, 24, and 48 h (Figure 7G). The malondialdehyde (MDA) content in *P. ussuriensis* gradually increased from 0 to 120 h and peaked at 120 h (Figure 7H).

### 2.10. Measurements of Minerals Content

In this study, we found that some genes related to ion transport were significantly different in expression after PEG treatment. Therefore, the mineral content (Ca, K, and Na) in the roots of *P. ussuriensis* was measured. The potassium (K), sodium (Na), and calcium (Ca) contents in the roots of *P. ussuriensis* were measured using inductively coupled plasma optical emission spectrometry (ICP-OES). The K and Ca contents in the roots of *P. ussuriensis* were significantly increased after PEG-induced drought stress and maintained at a high level at 120 h (Figure 8A,C). However, during PEG-induced drought stress, the Na content in the roots of *P. ussuriensis* did not significantly increase at 6, 12, and 24 h and sharply increased at 120 h (Figure 8B). In addition, the K and Ca levels in the roots of *P. ussuriensis* were much higher than Na levels.

Confocal images of Fluo-4 AM (Biyuntian Biotechnology Co., Ltd., Shanghai, China) fluorescence showed an increase in the fluorescence intensity of meristematic cells in roots after PEG treatment, which suggested a higher free Ca^2+^ accumulation in PEG treatment groups (Figure 9A–F). In the control group, the Ca^2+^ accumulation in the root tip cells was mainly distributed on the cell wall and nucleus (Figure 9A). With the increase in time for PEG-induced drought stress treatment, the Ca^2+^ accumulation in the root tip cells gradually and evenly filled the cytoplasm.

## 3. Discussion

### 3.1. Characteristics of PacBio SMRT-seq and Analysis of lncRNAs and AS Events

The PacBio SMRT-seq can generate long-read or full-length transcripts without splicing fragments. The full-length transcripts obtained from the PacBio SMRT-seq can be used for the identification and analysis of AS events and lncRNAs. The lncRNAs refer to RNA molecules whose transcripts are longer than 200 nt and do not encode proteins. The lncRNAs could function as competing endogenous RNAs (ceRNAs) to participate in response to drought stress in plants, and represent a new gene regulation layer [17,18]. In this study, we identified lncRNAs in *P. ussuriensis* under PEG-induced drought stress. However, their functions need to be further studied. Some mRNA precursors of eukaryotic genes produce different mRNA splicing isoforms through different splicing methods, which produce proteome and genetic diversity in eukaryotes [19,20]. There have been reports that AS mediates drought response, for example, maize plants overexpressing three different *ZmCCA1* isoforms distinguished by AS events showed different tolerance to drought stress [21], and *BrSR45a* regulates drought stress response by influencing AS of target genes [22]. Although many AS events have been identified in this study, different functions of these AS events on drought stress need to be further investigated. Therefore, in the future, research on the functions of these lncRNAs and AS events will help to better understand the response of *P. ussuriensis* to drought stress at the molecular level.

### 3.2. Sensing and Transmission of Drought Signals in P. ussuriensis

The drought stress perception and signal transduction in plants is a complex process. Plant cells sense drought stress signals through changes in the cell surface or cell membrane. After sensing the stress signal, plant cells stimulate and induce changes in the concentration of Ca^2+^, and then use calcium effector protein to transmit the signals [9,23]. In this study, the Ca^2+^ content in the roots of *P. ussuriensis* significantly increased after PEG-induced drought stress (Figure 8C and Figure 9). At the gene expression level, the genes encoding Ca^2+^ transporters (e.g., *ACA*, *CAX*, and *CNGC2* homologous genes) were up-regulated after drought stress. The results were consistent with that of the previous study that reported the contribution of *AtCNGC2* to Ca^2+^ uptake in *A. thaliana* [24]. Furthermore, we found that the genes encoding CML, calmodulin-binding protein, and CPK/CDPK homologous genes in the roots of *P. ussuriensis* were differentially expressed after PEG-induced drought stress. Notably, as calcium effector protein, most of these genes were only significantly up-regulated at 6 h and 12 h of PEG-induced drought stress. These results indicated that the transmission of Ca^2+^ signal mainly occurs in the early stages of drought stress.

Previous studies reported that ABA signaling plays an important role in plant acclimation to drought stress [9]. Zeaxanthin epoxidase (ZEP) is an important enzyme for the zeaxanthin epoxidation reaction in the ABA biosynthesis pathway, and overexpression of *AtZEP* enhanced the plant’s tolerance to drought stress [25]. NCED is the rate-limiting enzyme of the ABA biosynthesis pathway, and the *AtNCED3* gene is highly induced under drought stress [26]. In the roots of *P. ussuriensis*, one ZEP homologous gene and one *NCED3* homologous gene were up-regulated under PEG-induced drought stress, and their expression levels reached the highest value at 12 h of PEG-induced drought stress. This result was consistent with results in *Gossypium* spp. and *Vitis riparia* that NCED homologous genes were up-regulated under drought stress, suggesting that ABA signaling plays a role in plant responses to drought stress [9,27]. The ABA receptor PYR/PYL/RCAR, the negative regulator PP2C and the positive regulator SnRK2 are the main members in the ABA signaling pathway [28]. During PEG-induced drought stress, four *PYL* homologous genes were down-regulated and two *PP2C* homologous genes were up-regulated in the roots of *P. ussuriensis*. However, the details of the regulation of PYL–PP2C–SnRK2 in *P. ussuriensis* under drought stress are unknown.

### 3.3. Antioxidant Defense against Mechanism in Response to Drought Stress

ROS is a byproduct of plant aerobic metabolism, and excessive accumulation of ROS can damage the structure and function of plant cells [29,30]. In this study, after PEG-induced drought stress, the H_2_O_2_ content in the roots of *P. ussuriensis* was significantly increased (Figure 7A), and the count of MDA was also significantly increased (Figure 7H). These results indicated that the excessive accumulation of ROS might be one of the factors that cause severe damage to the cell membrane of *P. ussuriensis* under drought stress. Under abiotic stress, plants control the overproduction of ROS through enzymatic components and non-enzymatic antioxidants [29,30]. During drought stress, relatively high activity of enzymes (SOD, POD, CAT, and GST), which decompose ROS to protect cells from oxidative damage, was observed in the roots of *P. ussuriensis* (Figure 7B–E). The differential expression of ROS scavenging-related genes was observed in the roots of *P. ussuriensis* after PEG-induced drought stress (Appendix A), which provides evidence at the molecular level. Among them, more ROS clearance-related genes expression levels were significantly changed in the early stages (6, 12 and 24 h) of PEG-induced drought stress. These results on ROS in *P. ussuriensis* under drought stress are consistent with that of research reports on *Pinus halepensis*, *Malus pumila*, and *Hevea brasiliensis* [31,32,33].

### 3.4. Role of TFs in P. ussuriensis in Response to Drought Stress

TFs play a role in regulating the activation or inactivation of drought-responsive genes, which greatly affects the response of plants to drought stress [34,35]. In this study, the number of significantly differentially expressed TFs was highest at 6 h of PEG-induced drought stress, and gradually decreased along with the progressive drought (Appendix A). This result indicated that most TFs play a role in the early stages of drought stress. Similar results were obtained in *A. thaliana*, *O. sativa*, and *P. davidiana*, indicating that TFs are mainly involved in drought stress responses at 6 h and 12 h [8,36,37]. In this study, most of the differentially expressed TFs were members of the AP2/ERF, C2H2, MYB, NAC, C2C2, and WRKY TF families (Figure 5A). A substantial amount of evidence has shown that these TF subfamilies were involved in plant drought resistance [12,38,39,40].

Some TF family members play an important role in drought resistance through ABA-mediated pathways [10,41]. In-depth study of the WRKY TF family in response to drought stress has been carried out in many species. For example, overexpression of *GsWRKY20* in *A. thaliana* enhanced the drought tolerance by mediated ABA signaling, and overexpression of *BdWRKY36* in tobacco enhanced the drought tolerance and ABA biosynthesis [10,41]. In the present study, 10 WRKY TF homologous genes were differentially expressed, suggesting a specific role for these WRKY TFs in drought tolerance. MYB is one of the largest plant TF families, which positively modulates drought stress through ABA-mediated pathways. The overexpression of *SiMYB75* in *A. Thaliana* improved the plant’s tolerance to drought stress. Further analysis showed that ABA content was increased, and ABA-related genes were differentially expressed in SiMYB75-overexpressing lines [42]. Fang et al. [43] observed that overexpression of *PtrMYB94* in *Populus* enhanced the drought tolerance and increased the ABA content, suggesting that *PtrMYB94* is involved in the regulation of ABA-dependent drought stress in *Populus*. Several MYB family genes were differentially expressed at 6 h of PEG-induced drought stress, suggesting a regulatory role of MYB in the early stages of drought stress in *P. ussuriensis*.

Furthermore, in this study, we identified some AP2/ERF and NAC TF family members whose expression levels were significantly altered by PEG-induced drought stress. Similar results were obtained with other plants, such as *V. riparia*, *P. davidiana* and *L. kaempferi* [8,27,38]. Furthermore, the genes encoding *DRE1D*, *ERF5*, *NAC2*, and *NAC72*/*RD26* were up-regulated under PEG-induced drought stress in this study. Additionally, these TFs have been shown to have a positive regulation on drought resistance in other plants [44,45,46,47]. Furthermore, the genes that were down-regulated during drought stress also contributed to the adaptation of plants to drought stress. For example, Dong and Liu [48] observed that AtRAP2.1 negatively regulated drought stress in *A. thaliana*. Moreover, overexpression of *PeNAC034* in *A. thaliana* reduced drought tolerance, and *PeNAC045* overexpression line of *Populus* showed a drought-sensitive behavior [49]. In this study, we found that three *RAP* homologous genes and five *NAC* homologous genes were specifically down-regulated during PEG-induced drought stress in the roots of *P. ussuriensis* (Appendix A). Further exploration of these TFs may provide a greater understanding of drought resistance regulation in *P. ussuriensis*.

### 3.5. Ion Accumulation in P. ussuriensis

During drought stress, the osmotic balance in plant cells is destroyed, and water uptake is disturbed [50]. The changes in the ions in plant cells contribute to osmotic regulation. One unexpected finding of this study was the extent to which Ca^2+^ content increased significantly in cells at 48 and 120 h of PEG-induced drought stress (Figure 9). In addition, one *CNGC2* homologous gene was up-regulated at all time points of drought stress (Table 2), which mediates the absorption and accumulation of Ca^2+^ [51]. These results indicated that, in addition to acting as a second messenger, Ca^2+^ might also play an important role in osmotic regulation. Studies in *A. thaliana* have shown that Ca^2+^ contributed to the conversion of water channels from the active to an inactive state, thereby reducing the water permeability of the cytoplasmic membrane of the *A. thaliana* roots [52].

Prior studies have noted the importance of K^+^ in enhancing osmotic regulation in plants [53]. In this study, the K content in the roots of *P. ussuriensis* was significantly increased, and two *HAK/KT* homologous genes were significantly differentially expressed under PEG-induced drought stress (Table 2). Some transporters, namely *HvHAK1*, *HvAKT2*, *OsHAK1*, and *OsHAK5*, have been shown to be involved in the uptake of K^+^ in the roots [53,54]. Expression analysis in *Cajanus cajan* revealed that several K^+^ transport genes, that is, *CcHAK6*, *CcHAK15*, *CcAKT2*, and *CcHKT1*, were up-regulated in response to drought stress [55]. Notably, the K content in the roots of *P. ussuriensis* was significantly higher than that of Na and Ca under PEG-induced drought stress (Figure 9A–C). The results showed that the increase in K^+^ absorption might be the first choice for osmotic regulation in *P. ussuriensis*, under drought stress. Under PEG-induced drought stress, the Na content was significantly elevated in the roots of *P. ussuriensis*, but it was much lower than the K content levels. These results indicated that K^+^/Na^+^/Ca^2+^ accumulates in the roots of *P. ussuriensis*, which may contribute to the osmotic adjustment of the roots and limit the dehydration caused by PEG-induced drought stress.

### 3.6. Proteins, Osmolytes and Lignin Response to Drought Stress in P. ussuriensis

Along with antioxidant and ion accumulation, accumulation of proteins, osmolytes (such as proline and soluble sugars), and lignin plays an important role in drought stress. LEA and HSP proteins are important drought-inducing proteins in plants. LEA proteins belong to a large protein family of hydrophilic proteins that are closely associated with resistance to drought stress [16,56]. In-depth studies of the LEA protein family in response to drought stress have been carried out in other plants. For example, transgenic rice lines overexpressing *OsLEA3-1* have enhanced drought resistance compared to wild-type [57], and overexpression of *IpLEA* confers greater drought tolerance in *A. thaliana* [58]. In this study, we identified some of the *LEA* genes that were differentially expressed in the roots of *P. ussuriensis* under PEG-induced drought stress, especially five dehydrin genes that were up-regulated at all time points of PEG-induced drought stress (Appendix A). Furthermore, most of the DEGs encoding HSPs were up-regulated in the roots of *P. ussuriensis* under PEG-induced drought stress. The result is consistent with previous studies on *Capsicum annuum*, *O. sativa*, and *Solanum lycopersicum* showing that HSPs positively regulate the tolerance to drought stress [11,15,59].

The essential role of osmolytes is to limit water loss in plants under drought stress. Proline is a common plant organic osmotic agent, which is closely related to drought stress [50]. Proline content increased under drought stress, which has been reported in *A. thaliana*, *Populus euphratica*, and *Populus yunnanensis* [60,61,62]. One of the most common permeates in plants is soluble sugar, and the accumulation of total soluble sugar contributes directly to drought tolerance. For example, drought significantly induced the accumulation of total soluble sugar in *Michelia macclurei* and *Schima superba* [63]. In addition, Regier et al. [64] observed that total soluble sugar content increased in the roots of water-limited *Populus nigra* compared to well-watered *P. nigra*. Our results clearly showed that the proline content and total soluble sugar content of *P. ussuriensis* were significantly increased in response to PEG-induced drought stress (Figure 7F,G). In addition, the amino acid biosynthesis; cysteine and methionine metabolism; and glycine, serine, and threonine metabolism pathways were significantly enriched at 6 and 12 h of PEG-induced drought stress (Figure 4). The results suggested that proline and total soluble sugar levels positively regulated the drought stress tolerance in *P.*
*ussuriensis*.

As the main component of the secondary cell wall, lignin plays an important role in plant resistance to drought stress [65,66]. Recently, it has been demonstrated that the increase in lignin enhanced the tolerance of *P. ussuriensis* to drought stress [67]. The lignin monomers were carried out through the phenylpropanoid biosynthesis pathway. Moreover, in the present study, we found that the phenylpropanoid biosynthesis pathway was significantly enriched under PEG-induced drought stress (Figure 4). Increased expression of genes involved in lignin biosynthesis, such as *PAL*, *4CL*, *CAD*, *CCR*, *C4H*, and *COMT* homologous genes, was observed in the roots of *P. ussuriensis* under PEG-induced drought stress (Appendix A). Similar results were obtained in *Paeonia ostii* [68] and *L. kaempferi* [38], which suggested that plants might promote lignin biosynthesis resulting in enhanced tolerance to drought stress.

## 4. Materials and Methods

### 4.1. Plant Material and Drought Treatment

In this study, the tissue culture seedlings of *P. ussuriensis* were used as research material. The stem (3-cm long) of *P. ussuriensis* was cut and cultured in half-strength Murashige and Skoog (MS) medium for rooting and grown in a tissue-culture chamber (temperature 25 ± 2 °C, photoperiod of 16 h light and 8 h dark). After three weeks, the tissue culture seedlings of *P. ussuriensis* had a complete root development (Appendix A). Three-week-old *P. ussuriensis* seedlings with uniform root growth were then transferred to liquid media containing 6% PEG 6000 (Sigma–Aldrich, St. Louis, MO, USA). After treatment of 0, 6, 12, 24, 48, and 120 h, the roots were harvested at the same time and frozen in liquid nitrogen. Each sample was from at least six different seedlings, and three biological replicates were collected at each treatment.

### 4.2. Preparation of Full-Length cDNA Library and Illumina RNA-seq Library and Sequencing

Total RNA was isolated as described previously [69], with a slight modification. Then, RNA integrity was assessed according to the requirement of 28S:18S ≥ 1.4, RNA Integrity Number (RIN) ≥ 8. The total RNA from 18 samples was combined in equal amounts for cDNA synthesis and SMRTbell library construction. The cDNA synthesis was performed using the Clontech SMARTer PCR cDNA synthesis kit (Takara Biotechnology, Dalian, China), and the SMRTbell library was constructed using the PacBio SMRTbell Template Prep Kit (PacBio, Menlo Park, CA, USA). In total, 2 μg of RNA was used to generate sequencing libraries using the NEBNext Ultra RNA Library Prep Kit (#E7530L, NEB, Ipswich, MA, USA). Libraries were sequenced in 150 b.p. paired-end mode, using an Illumina HiSeq X Ten system. The above sequencing was performed by Zhejiang Annoroad Biotechnology Co., Ltd. (Beijing, China).

### 4.3. Quality Control and Error Correction of PacBio SMRT-seq and RNA-seq

Quality control of the raw data was performed to avoid inaccuracy in subsequent information analysis. After data quality control, the data volume of subreads was calculated. The consensus sequence generated by multiple subread sequences are CCS reads, and the CCS reads are further corrected to obtain HQ isoforms (with accuracy greater than 0.99) and LQ isoforms. The isoforms with a large number of redundant sequences were clustered together using an isoform-level clustering algorithm known as iterative clustering for error correction (ICE) to obtain a new consensus isoform [70]. The consensus isoform obtained after ICE was used as the reference sequence, and the short-read sequences obtained from 18 Illumina RNA-seq libraries were compared to the reference sequence using RNA-seq by Expectation Maximization (RSEM) (v1.2.31) software (Li, Dewey, WI, USA) [71].

### 4.4. SSR and lncRNA Prediction

The SSRs were identified using the MISA software [72]. The lncRNAs were predicted using CNCI [73], CPC [74], and CPAT [75].

### 4.5. Identification of Alternative Splicing Events

In AS, different splicing of mRNA precursors can produce different mRNA splicing isoforms [19,20]. In this study, AS events were predicted using AStalavista 4.0 (http://confluence.sammeth.net/display/ASTA/Home, accessed on 1 August 2021) [76].

### 4.6. Transcript Integrity Analysis and Gene Annotation

In this study, the integrity of transcripts generated by PacBio SMRT-seq was analyzed using benchmarking universal single-copy orthologs (BUSCO) [77]. The ORFs were predicted using TransDecoder version 3.0.1 [78]. Then, the functional annotations of predicted ORFs were combined using Trinotate, which uses multiple well-referenced methods for functional annotation.

### 4.7. Analysis of Differentially Expressed Transcripts

Using the results from the PacBio SMRT-seq as a reference, the differential gene expression analysis between different treatment groups was performed using the DEGSeq software [79]. Genes satisfying the conditions of |log2 fold change rate| ≥ 1 and *p*-value < 0.05 were identified as DEGs. The visualization of the expression patterns of DEGs is achieved by building a heatmap using the pheatmap package (https://cran.r-project.org/web/packages/pheatmap, accessed on 6 April 2021). GO enrichment analysis and KEGG pathway enrichment analysis of DEGs using GOseq R package and KOBAS software, respectively [80,81].

### 4.8. Validation of Gene Expression Levels

A total of 12 genes of differential expression were selected for validation of results of RNA-seq by qRT-PCR. These 12 DEGs were *CML38* (f2p30_1011_11035), *MAPKKK18* (f5p60_1300_26955), *NCED3* (f2p21_2499_2742), *KT2* (f7p60_2856_19557), *NCL* (f2p60_2293_3245), *CAX7* (f3p60_1980_22126), *RAB18* (f2p60_1369_7946), *LEA29* (f2p37_1282_27056), *DNJ11* (f2p60_820_32228), *PER4* (f2p59_1311_26810), *GSTXC* (f2p60_894_312621), and *SODC* (f2p28_837_12827) homologous genes. The polyubiquitin 10 (*UBQ10*) gene was used as an internal control. The primers of the 12 DEGs are listed in Appendix A. For accuracy, three technical replicates were used for each qRT-PCR reaction. The relative abundance of each transcript was determined using the 2^−ΔΔCT^ method.

### 4.9. Determination of Physiological Indexes

The MDA content was measured using a plant MDA assay kit. The total soluble sugar content was measured using a plant soluble sugar content test kit. The proline content was measured using a proline assay kit. The CAT activity was measured using a CAT assay kit. The GST activity was measured using a GST assay kit. The SOD activity was measured using a total SOD assay kit. The POD activity was measured using a POD assay kit. All assay kits were purchased from Nanjing Jiancheng Bioengineering Institute, China. These data were averaged from three biological replicates.

### 4.10. Measurements of Minerals Content

The root samples were taken in triplicate at 0, 6, 12, 24, 48, and 120 h after PEG-induced drought stress. The whole roots were dried using an oven. The minerals content (Ca, K, and Na) in the roots of *P. ussuriensis* was assayed using ICP-OES, as described previously, with slight modifications [82]. These data were averaged from three biological replicates. In addition, according to the protocol outlined by Li et al. [14], the free Ca^2+^ content in root tip cells was measured using calcium indicator Flou-4 AM (Biyuntian, Shanghai, China), and observed through a confocal laser scanning microscope imager 700 (Carl Zeiss, Docuval, Germany).

### 4.11. Statistical Analysis of Data

In this study, statistical analysis of data was performed using statistical package for social sciences (SPSS) version 20 (IBM, Chicago, IL, USA), and analysis of significance was performed using analysis of variance (ANOVA). In addition, correction for multiple testing using the Bonferroni approach. Significant differences in relative levels were represented by different lowercase letters, *p* < 0.05.

## 5. Conclusions

The purpose of the current study was to determine the responses of *P. ussuriensis* root to drought stress using physiochemical measurements, SMRT-seq and RNA-seq analysis. A hypothetical model of the drought tolerance mechanism for *P. ussuriensis* was established by integrating the physiological and molecular responses as well as plant growth under drought stress (Appendix A). The findings clearly indicate that a series of potential drought-responsive genes related to signal transduction, antioxidant, ion accumulation and osmotic regulation in the roots of *P. ussuriensis* were differentially expressed under drought stress. Combined with the physiological and biochemical indexes, we found the ion accumulation, the content of proline and soluble sugar, antioxidase activity (SOD, CAT, POD) and non-enzymatic antioxidant (GST) activity were all changed in *P. ussuriensis* root during PEG-induced drought stress. In summary, the combined analysis of temporal transcriptomic and physiological revealed complex genes and networks involved in the response of *P. ussuriensis* roots to different stages of drought stress.

## Figures and Tables

**Figure 1 ijms-23-03840-f001:**
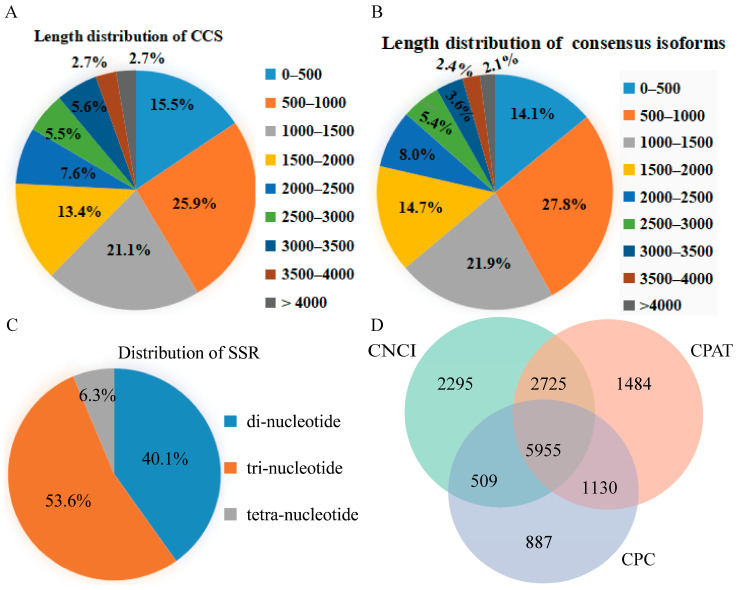
(**A**) Length distribution of circular consensus sequencing (CCS) reads, (**B**) length distribution of isoforms, (**C**) type distribution of simple sequence repeats (SSRs), and (**D**) Venn diagram of the predicted long non-coding RNAs (lncRNAs).

**Figure 2 ijms-23-03840-f002:**
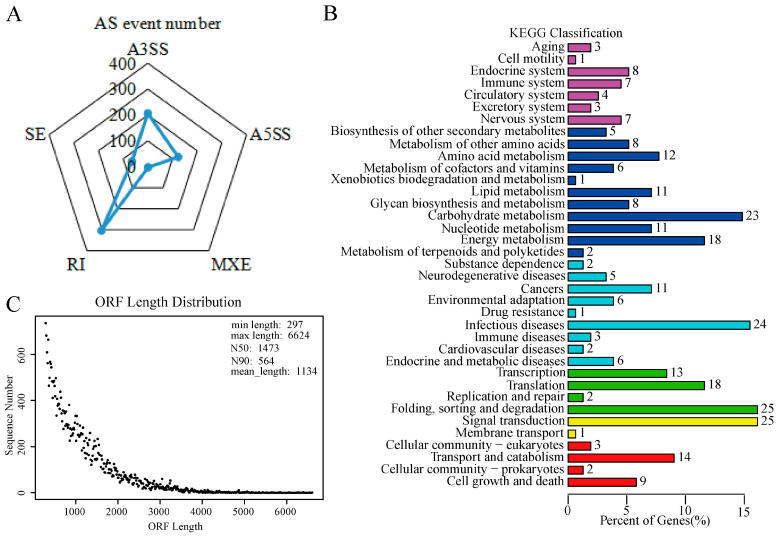
Alternative splicing (AS) identification and analysis, and open reading frame (ORF) prediction. (**A**) Statistics of the distinct AS events, (**B**) KEGG enrichment analysis of AS genes, and (**C**) length distribution of ORFs.

**Figure 3 ijms-23-03840-f003:**
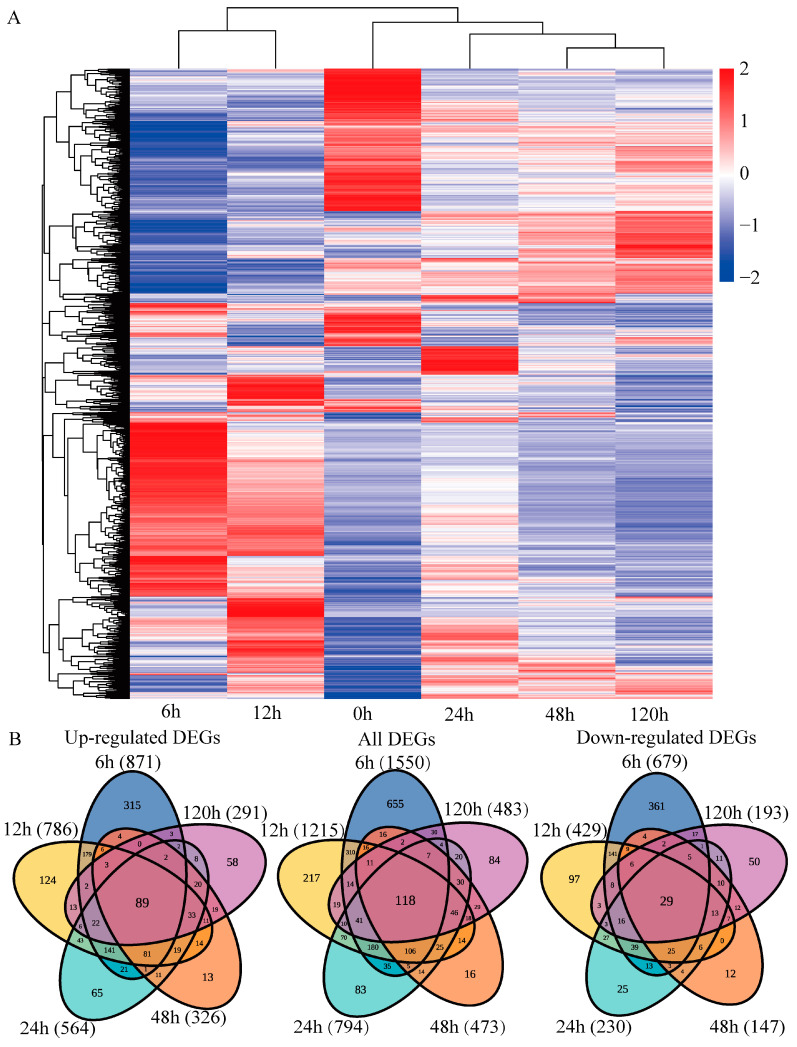
Overview of the differentially expressed genes (DEGs). (**A**) Clustering analysis and (**B**) Venn diagrams of DEGs.

**Figure 4 ijms-23-03840-f004:**
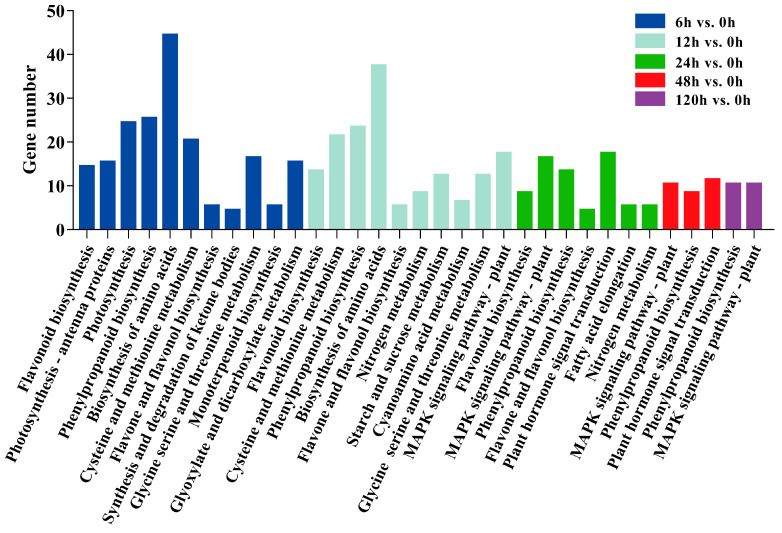
KEGG enrichment analysis of DEGs.

**Figure 5 ijms-23-03840-f005:**
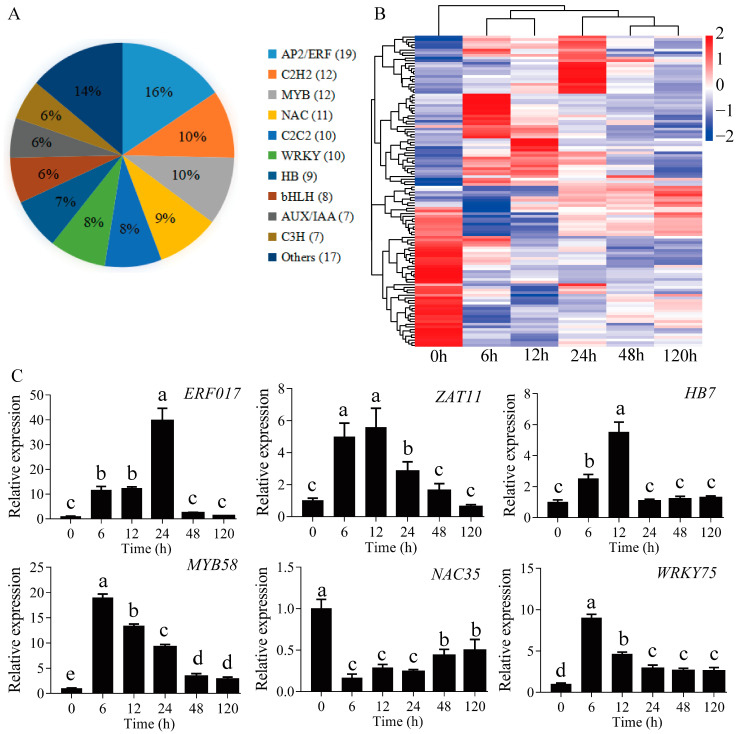
Transcription factor analysis. (**A**) Distribution of transcription factors sorted into different families, (**B**) clustering analysis of transcription factors, and (**C**) qRT-PCR of different transcription factors. Different lowercase letters indicate statistically significant differences at *p* < 0.05. The values are shown as the mean of three replicates ± SE (*n* = 3).

**Figure 6 ijms-23-03840-f006:**
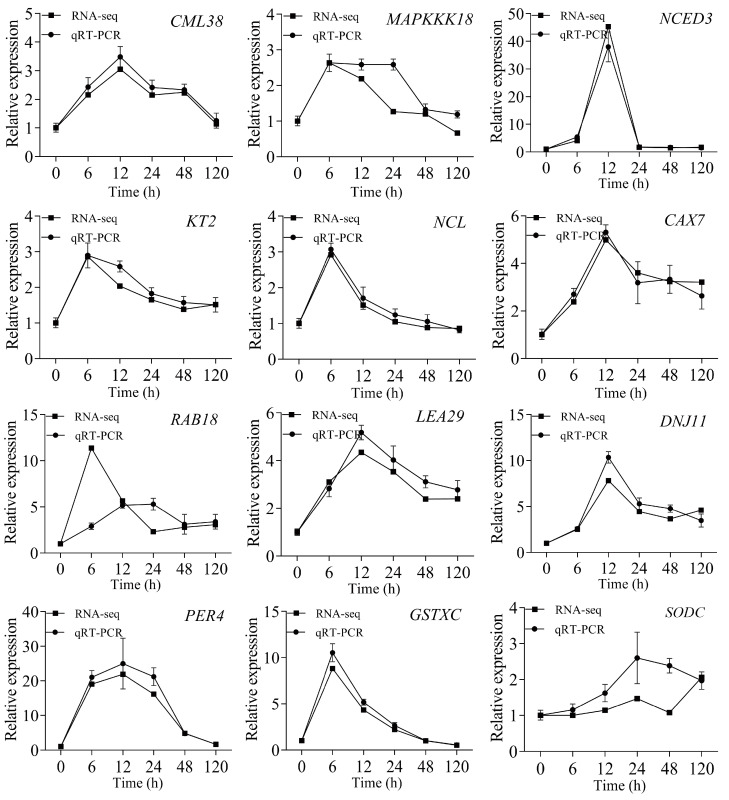
Validation of gene expression levels. The values are shown as the mean of three replicates ± SE (*n* = 3).

**Figure 7 ijms-23-03840-f007:**
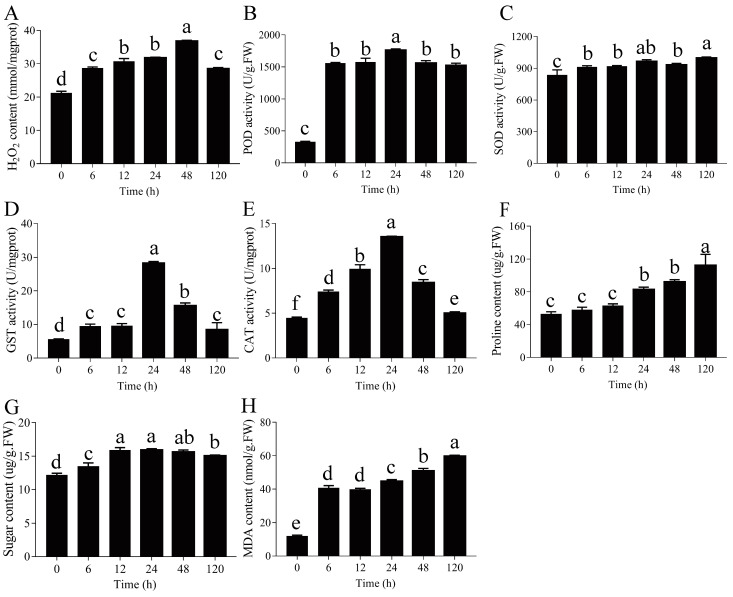
Effects of drought stress on the roots of *Populus ussuriensis*. (**A**) H_2_O_2_ content, (**B**) peroxidase (POD) activity, (**C**) superoxide dismutase (SOD) activity, (**D**) glutathione S-transferase (GST) activity, (**E**) catalase (CAT) activity, (**F**) proline content, (**G**) total soluble sugar content, and (**H**) malondialdehyde (MDA) content. Different lowercase letters indicate statistically significant differences at *p* < 0.05. The values are shown as the mean of three biological replicates ± SE (*n* = 3).

**Figure 8 ijms-23-03840-f008:**
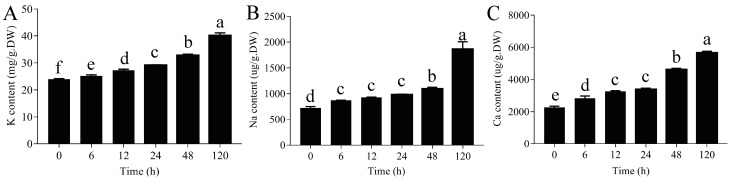
Measurement of mineral content. (**A**) K content, (**B**) Na content, and (**C**) Ca content. Different lowercase letters indicate statistically significant differences at *p* < 0.05. The values are shown as the mean of three biological replicates ± SE (*n* = 3).

**Figure 9 ijms-23-03840-f009:**
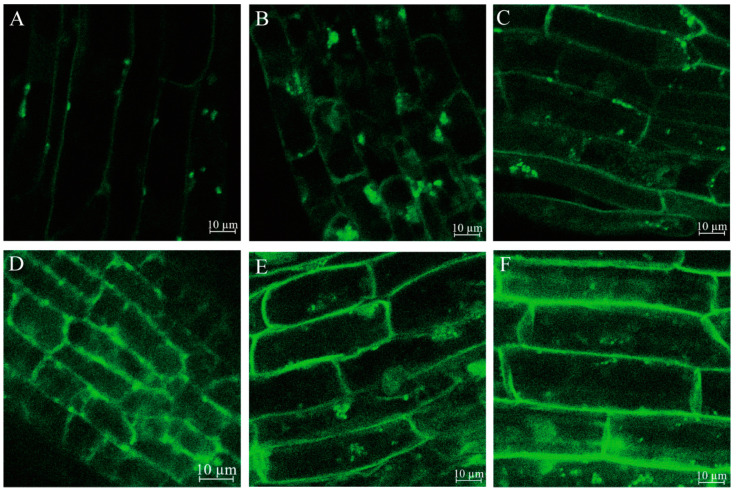
Calcium imaging in different treatments. (**A**) Control treatment, (**B**) 6 h post 6% PEG treatment, (**C**) 12 h post 6% PEG treatment, (**D**) 24 h post 6% PEG treatment, (**E**) 48 h post 6% PEG treatment, and (**F**) 120 h post 6% PEG treatment. The scale bar is 10 μm.

**Table 1 ijms-23-03840-t001:** Overview of the Single-Molecule Real-Time sequencing in *Populus ussuriensis*.

Name	Reads Number	Base Number	Average_Length
Polymerase reads	419,372	33,074,826,044	78,867.51
Subreads	31,547,319	29,488,228,404	943.73
CCS reads	387,340	566,167,282	1461.68
High-quality Isoforms	40,307	56,721,399	1407.23
Low-quality Isoforms	152	128,876	847.87

**Table 2 ijms-23-03840-t002:** List (sections) of differentially expressed genes in drought.

Gene ID	Annotation	Log2 Value of Fold Change				
		6 h vs. 0 h	12 h vs. 0 h	24 h vs. 0 h	48 h vs. 0 h	120 h vs. 0 h
Signaling						
*f2p60_1754_23479*	Protein phosphatase 2C 3	2.01	5.50	0.75	0.57	0.77
*f2p60_653_34458*	Cytochrome P450, family 707, (CYP707A1)	2.05	3.19	1.80	1.90	1.75
*f2p60_1246_27481*	Abscisic acid receptor PYL4	−1.14	−1.68	−1.22	−1.32	−1.04
*f2p60_745_332685*	Calcium-binding protein PBP1	−1.87	−1.51	−2.29	−2.18	−3.70
*f2p60_1346_26477*	Phosphoenolpyruvate carboxylase kinase 2	−2.06	−2.35	−2.20	−1.47	−1.04
*f2p60_482_36687*	Calcium-dependent protein kinase 1	−0.67	−2.34	−1.67	−0.35	−0.70
*f2p30_1011_11035*	Calcium-binding protein CML38	0.60	2.42	0.90	1.72	1.39
*f2p60_490_36587*	Calmodulin-like protein 5	1.11	1.61	1.11	1.17	0.18
*f4p60_1325_8326*	Mitogen-activated protein kinase kinase kinase 18	0.48	1.59	0.98	1.27	1.84
Ion transport						
*f2p37_2396_20662*	Cyclic nucleotide-gated ion channel 2	0.81	1.01	2.58	1.10	−0.07
*f2p60_2781_19668*	Potassium transporter 5	0.38	1.29	0.85	1.15	1.39
*f7p60_2856_19557*	Potassium transporter 2	−1.73	−2.50	−0.97	−0.47	0.20
*f3p60_1980_22126*	Calcium exchanger 7 (CAX7)	1.52	1.03	0.73	0.47	0.60
*f4p60_1851_22830*	Autoinhibited Ca(2+)-ATPase 10	1.25	2.32	1.85	1.70	1.68
*f2p60_2293_3245*	Sodium/calcium exchanger	−2.82	−0.92	−0.02	−0.07	0.09
*f4p60_2032_21887*	WRKY transcription factor 33	1.10	0.99	0.39	0.17	0.27
*f2p57_4009_433*	Calcium-transporting ATPase 10	2.29	1.24	0.64	0.55	0.49
*f2p55_3201_1361*	Glutamate receptor 2.7	−2.99	−1.58	−2.36	−1.78	−0.74
*f2p60_3228_18926*	Glutamate receptor 3.6	0.33	0.91	0.77	1.21	1.41
*f2p20_3494_18605*	Calcium-transporting ATPase 13	0.93	0.73	−0.26	−0.43	−1.30
Reactive oxygen species						
*f2p60_1104_10166*	Glutathione S-transferase L3	1.54	0.60	0.06	−0.18	−0.22
*f3p60_1525_6924*	Glutathione S-transferase L3-like	2.43	2.07	1.23	0.73	0.73
*f2p60_894_312621*	Glutathione S-transferase parC	2.51	2.03	1.22	0.74	0.53
*f9p60_892_31310*	Glutathione S-transferase parC	1.99	1.39	0.57	−0.32	−1.30
*f2p40_1265_8944*	Glutathione S-transferase L3	1.97	1.48	1.06	0.69	0.43
*f3p60_913_11828*	Glutathione S-transferase parA	1.87	1.02	0.43	0.54	−0.08
*f2p60_993_30050*	Glutathione S-transferase parC	1.82	0.80	0.07	0.24	−0.07
*f2p60_1054_29478*	Glutathione S-transferase 6	1.12	1.88	0.14	0.19	−0.61
*f3p60_1090_29044*	Glutathione S-transferase 4	0.14	0.12	2.03	0.28	−0.18
*f2p59_1311_26810*	Peroxidase 4	3.14	2.11	1.13	−0.03	−0.97
*f4p60_1269_8729*	Peroxidase 5	4.25	4.45	4.01	2.27	0.70
Lignin biosynthetic process						
*f2p60_2079_21713*	4-coumarate-CoA ligase 1	−0.63	−1.42	−0.42	−2.26	−5.21
*f3p60_2144_3723*	Phenylalanine ammonia-lyase 2	2.79	2.51	2.10	0.99	0.56
*f3p60_2656_2364*	Phenylalanine ammonia-lyase G2B	2.40	2.02	1.28	1.08	0.62
*f2p60_2638_2450*	Phenylalanine ammonia-lyase G2B	2.11	2.05	1.61	1.15	0.91
*f2p60_1840_22893*	Trans-cinnamate 4-monooxygenase	2.25	2.05	1.43	0.84	0.64
*f2p60_1499_25217*	Cinnamyl alcohol dehydrogenase 1	2.16	1.99	1.35	0.79	0.25
*f6p60_1412_25819*	Cinnamyl alcohol dehydrogenase 9	2.70	2.13	1.05	0.34	−0.39
*f2p60_1466_7341*	Cinnamoyl-CoA reductase 1	0.54	1.81	1.88	1.75	1.42
Proteins						
*f3p60_839_12609*	18.5 kDa class I heat shock protein	1.64	1.77	1.61	1.22	0.87
*f3p60_787_32671*	17.3 kDa class I heat shock protein	2.58	3.24	1.54	1.45	0.85
*f3p60_885_12148*	22.7 kDa class IV heat shock protein	2.88	3.24	1.61	1.14	0.35
*f3p60_805_32408*	18.2 kDa class I heat shock protein	2.82	3.08	1.03	0.70	−0.18
*f4p60_842_31916*	17.9 kDa class II heat shock protein	2.51	2.56	1.26	1.08	0.20
*f2p60_797_32529*	17.6 kDa class I heat shock protein 3	2.12	2.43	0.96	0.34	−0.61
*f2p60_820_32228*	Chaperone protein dnaJ 11	2.26	2.41	0.83	0.94	0.55
*f2p60_1369_7946*	Dehydrin Rab18-like	1.34	2.97	2.16	1.87	2.20
*f8p60_825_32305*	Dehydrin family protein	3.51	2.50	1.21	1.48	1.61
*f13p60_601_14927*	Late embryogenesis abundant protein family protein	2.13	1.67	1.59	1.34	1.32
*f2p37_1282_27056*	Late embryogenesis abundant protein D-29	3.02	1.59	1.04	1.08	0.77
*f2p45_1593_6461*	Late embryogenesis abundant protein 4	1.64	2.12	1.82	1.26	1.26

## Data Availability

The raw sequence read from PacBio single-molecule real-time sequencing has been uploaded to the Sequence Read Archive database (SRA accession: PRJNA809518). The raw sequence read from Illumina RNA sequencing has been uploaded to the GEO database (GEO accession: GSE197096).

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
