# Peer review of "Characterization of the Gene Expression Profile Response to Drought Stress in Populus ussuriensis Using PacBio SMRT and Illumina Sequencing"

_ijms, 2022, doi:10.3390/ijms23073840_

Round 1

Reviewer 1 Report

1) The phrase " and Illumina sequencing" should be removed from the line 14 due to the duplication error.

2)  It's better to add N50 metrics when writing about statistics of PacBio SMRT-seq (lines 72-73, 75-76).

3) Did the authors conduct any completeness analysis for transcripts generated with PacBio SMRT-seq (e.g. BUSCO)? In addition to given statistics conserning the number and length of identified genes, such analysis could help to understand, how complete (to the biological point of view) the set of revealed transcripts is, and so is highly recommended.

4) The reviewer thinks that the plots showing consistency between transcriptomics and qRT-PCR results should be of the same type (as on the Figure 6, but with the addition of lowercase letters indicate statistically significant differences). In the given version of the paper there are no lowercase letters on the Figure 6, however they are promised in the figure caption.

5) Did the authors use any multiple testing corrections during statistical anylisis (section 4.11)? Such corrections are meant to reduce the probability of false-positive results in case of conduction of multiple pairwise comparisons, so are needed if the researchers have more than one pair of samples to compare.

Author Response

Thank you for taking the time to provide constructive comments on our manuscript. We have implemented comments and suggestions. Changes in the modified section are tracked and minor revisions have been made in the diagram. Below, we've also provided point-by-point responses explaining how we handle each comment. Comment 1 The phrase "and Illumina sequencing" should be deleted from line 14 due to duplication errors. Our response: Thanks for your suggestion, we have changed the sentence on page 1, line 11. 

Comment 2

It's better to add N50 metrics when writing about statistics of PacBio SMRT-seq (lines 72-73, 75-76).

Our response: Thank you for this suggestion, we have added the N50 metrics in lin page 2, lines 68-70.

Comment 3

Did the authors conduct any completeness analysis for transcripts generated with PacBio SMRT-seq (e.g. BUSCO)? In addition to given statistics conserning the number and length of identified genes, such analysis could help to understand, how complete (to the biological point of view) the set of revealed transcripts is, and so is highly recommended.

Our response: Thank you for this suggestion, we have assessed the integrity of transcripts generated by PacBio SMRT-seq using BUSCO, and added  method descriptions and citations in page 17, line 482.

Comment 4

The reviewer thinks that the plots showing consistency between transcriptomics and qRT-PCR results should be of the same type (as on the Figure 6, but with the addition of lowercase letters indicate statistically significant differences). In the given version of the paper there are no lowercase letters on the Figure 6, however they are promised in the figure caption.

Our response: Thank you for this suggestion. That was not our intention and we have decided to remove the sentence from figure caption for clarity.

Comment 5

Did the authors use any multiple testing corrections during statistical anylisis (section 4.11)? Such corrections are meant to reduce the probability of false-positive results in case of conduction of multiple pairwise comparisons, so are needed if the researchers have more than one pair of samples to compare.

Our response: Thank you for your suggestion. Statistical analysis was performed using one-way ANOVA followed by correction for multiple testing using the Bonferroni method. We added this sentence in Section 4.11 for clarity. 

Reviewer 2 Report

The manuscript is generally well written, though there are sections that require revisions to style and grammar. In particular there are section that are repeated (identical lines appear the the abstract and introduction. Some repetition of words such as "In plants, the root is the first organ to sense drought stress, and it plays a vital role in the response of plants to drought stress" 

Some figures are low resolution and could be improved for the final manuscript.

My primary concern is how representative the study is, which should be addressed by the authors.

1) The PEG treatment is short in duration, often under real world conditions drought can take months or years to deplete soil moisture. Is a 24hr 6% PEG treatment sufficient? Why was 6% solution employed and what level of osmotic stress does this represent? PEG screening is not an entirely perfect technique as it may limit oxygen uptake in solutions, please clarify some of the limitations and benefits associated with the methodology.   

2) The materials employed are three week old seedlings, do these represent typical materials for transplanting P. ussuriensis? Please clarify, my concern is if the materials have had sufficient time to develop root mass.

Author Response

We would like to express our thanks to you for your time and constructive comments on our manuscript. We have implemented the comments and suggestions. Changes were tracked for the modified parts and a minor modification was made in the figure. Below, we also provide a point-by-point response explaining how we have addressed each of the comments.

Comment 1

Some repetition of words such as "In plants, the root is the first organ to sense drought stress, and it plays a vital role in the response of plants to drought stress" 

Our response: Thank you for this suggestion. We have decided to remove the sentence from the abstract.

Comment 2

Some figures are low resolution and could be improved for the final manuscript.

Our response: Thank you for this suggestion. We have tried our best to improve the graph quality in the revised version, such as Figure 4.

Comment 3

The PEG treatment is short in duration, often under real world conditions drought can take months or years to deplete soil moisture. Is a 24hr 6% PEG treatment sufficient? Why was 6% solution employed and what level of osmotic stress does this represent? PEG screening is not an entirely perfect technique as it may limit oxygen uptake in solutions, please clarify some of the limitations and benefits associated with the methodology.

Our response: Thank you for your question. In this study, we characterized the gene expression profile in roots of Populus ussuriensis at 0, 6, 12, 24, 48 and 120 h after the start of drought stress, which is difficult to achieve by water control, so PEG6000 was used to simulate drought. Polyethylene glycol 6,000 (PEG 6000) is inert, non-ionic and cell impermeable. They are small enough to influence the osmotic pressure, but large enough not to be absorbed by plants. Therefore, they are frequently used to simulate drought stress. PEG facilitates quantitative design of experiments. PEG mimics drought stress is more suitable for short-term treatment to observe changes in gene expression. However, PEG cannot fully simulate drought, so it is not suitable for long-term drought stress experiments. In this study, the concentration of PEG was obtained by pre-experiment. For the tissue culture seedlings of Populus ussuriensis, the drought simulated by 6% PEG6000 stress was equivalent to moderate drought stress.

Comment 4

The materials employed are three week old seedlings, do these represent typical materials for transplanting P. ussuriensis? Please clarify, my concern is if the materials have had sufficient time to develop root mass.

Our response: Thank you for your question. The stem (3-cm long) of Populus ussuriensis was cut and cultured in half-strength Murashige and Skoog (MS) medium for rooting. After three weeks, the tissue culture seedlings of Populus ussuriensis had a complete root development. We have added the sentence and added Figure S5 in page 16, lines 443-445.

Reviewer 3 Report

In this manuscript, the author characterized the gene expression profile response to drought stress in Populus ussuriensis using PacBio SMRT and Illumina sequencing. In this research, authors characterized the gene expression profile in roots of Populus ussuriensis at 0, 6, 12, 24, 48, and 120 h after the start of polyethylene glycol (PEG)-induced drought stress using PacBio single-molecule real-time sequencing (SMRT-seq) and Illumina sequencing and Illumina RNA sequencing. Compared to the control, 2,244 differentially expressed genes (DEGs) were identified, and many of these DEGs were associated with the signal transduction, antioxidant system, ion accumulation, and drought-inducing proteins. Furthermore, changes in specific physiological and biochemical indexes, such as antioxidant activity and the contents of Ca2+, proline, and total soluble sugars, were further confirmed in P. ussuriensis roots. Furthermore, most of the differentially expressed transcription factors were members of the AP2/ERF, C2H2, MYB, NAC, C2C2, and WRKY families. Additionally, based on PacBio SMRT-seq results, 5,955 long non-coding RNAs and 700 alternative splicing events were identified. These results provide a global view of gene expression profile that contributes to drought resistance in P. ussuriensis and meaningful information for genetic engineering research in the future.

The manuscript has solid data but lacks refinement. Also, I have detected plagiarism in this study. Please find my detailed comments below.

At L33, change from water status, promotes to water status promotes.

L46 ABA responsive to ABA-responsive.

L49 protein, enhances to protein enhances.

L55 remain unexplored to remains unexplored.

L72.L73 change from bp to b.p.

L97 exon , to exon,.

L127 the change pattern to the changing pattern.

L128 PEG treatment, was to PEG treatment was.

L269 for exemple to for example.

L366 provide greater understanding to provide a greater understanding.

L379 Author have written upregulated 3 times and up-regulated 5 times. Please be consistant.

L377 active to inactive state to active to an inactive state.

From L510-518 What is leves?

I found plagiarism in this manuscript at L28, L45-46, L60-61, L74-75, L78-79, L95-97, L99-100, L149, L161-162, L175, L181-182, L197-198, L201-202, L261-262, L277, L302-303, L326, L422-423, L426, L432, L439, L440, L446-447, L452-462, L466-467, L469-472. Please clean.

Author Response

We would like to express our thanks to you for your time and constructive comments on our manuscript. We have implemented the comments and suggestions. Changes were tracked for the modified parts and a minor modification was made in the figure. Below, we also provide a point-by-point response explaining how we have addressed each of the comments.

Comment 1

At L33, change from water status, promotes to water status promotes.

Our response: Thank you for this suggestion. We have changed the sentences in page 1, line 31.

Comment 2

L46 ABA responsive to ABA-responsive.

Our response: Thank you for this suggestion. We have changed the sentences in page 1, line 42.

Comment 3

L49 protein, enhances to protein enhances.

Our response: Thank you for this suggestion. We have changed the sentences in page 1, line 45.

Comment 4

L55 remain unexplored to remains unexplored.

Our response: Thank you for this suggestion. We have changed the sentences in page 1, line 52.

Comment 5

L72.L73 change from bp to b.p.

Our response: Thank you for this suggestion. We have changed and checked the writing of “b.p.” throughout the MS.

Comment 6

L97 exon , to exon,.

Our response: Thank you for this suggestion. We have changed the sentences.

Comment 7

L127 the change pattern to the changing pattern.

Our response: Thank you for this suggestion. We have changed the sentences in the page 4, lines 125-126.

Comment 8

L128 PEG treatment, was to PEG treatment was.

Our response: Thank you for this suggestion. We have changed the sentences in the page 4, line 126.

Comment 9

L269 for exemple to for example.

Our response: Thank you for this suggestion. We have changed the sentences in the page 13, line 265.

Comment 10

L366 provide greater understanding to provide a greater understanding.

Our response: Thank you for this suggestion. We have changed the sentences in the page 15, line 366.

Comment 11

L379 Author have written upregulated 3 times and up-regulated 5 times. Please be consistant.

Our response: Thank you for this suggestion. We have changed and checked the writing of “up-regulated” throughout the MS.

Comment 12

L377 active to inactive state to active to an inactive state.

Our response: Thank you for this suggestion. We have changed the sentences in the page 15, line 376.

Comment 13

From L510-518 What is leves?

Our response: Thank you for your question. It's our writing error, the levels mean is content leaves, we have changed the leves to content for clarity. 

Comment 14

I found plagiarism in this manuscript at L28, L45-46, L60-61, L74-75, L78-79, L95-97, L99-100, L149, L161-162, L175, L181-182, L197-198, L201-202, L261-262, L277, L302-303, L326, L422-423, L426, L432, L439, L440, L446-447, L452-462, L466-467, L469-472. Please clean.

Our response: :Thank you for this suggestion. These section was reworded to avoid potential issues of plagiarism. In addition, we checked the full text for plagiarism and revised.

Round 2

Reviewer 3 Report

I am happy with the author's comments and the manuscript looks refined now. It can be accepted after minor changes.

  1. Make gene id italic in table 2.
  2. Make gene name italic in figure 5c and 6.
  3. If possible make one hypothetical figure to depict finding of this study.

Author Response

We would like to express our thanks to you for your time and constructive comments on our manuscript. We have implemented your comments and suggestions and wish to submit a revised version of the manuscript for further consideration in the journal. Changes were tracked for the modified parts and a minor modification was made in the figure. Below, we also provide a point-by-point response explaining how we have addressed each of the  comments.

Comment 1 

Make gene id italic in table 2.

Our response: Thank you for this suggestion, we have maked the gene id italic in table 2.

Comment 2

Make gene name italic in figure 5c and 6.

Our response: Thank you for this suggestion, we have changed the figure 5c and figure 6.

Comment 3

If possible make one hypothetical figure to depict finding of this study.

Our response: Thank you for this suggestion, we have added one hypothetical model of the drought tolerance mechanism for P. ussuriensis (Figure S6) in page 18, line 534.